# Modelling Nitrogen Uptake in Plants and Phytoplankton: Advantages of Integrating Flexibility into the Spatial and Temporal Dynamics of Nitrate Absorption

**Erwan Le Deunff [1],\*, Philippe Malagoli [2]**  **and Marie-Laure Decau [3]**

1. ICORE Structure Fédérative Interactions Cellules ORganismes Environnement, Normandie Université, UNICAEN, ICORE, F-14000 Caen, France
2. Université Clermont Auvergne, INRA, PIAF, F-63000 Clermont-Ferrand, France; philippe.malagoli@uca.fr
3. INRA Unité Expérimentale Fourrages Environnement Ruminants (FERLUS) et Système d'Observation et d'Expérimentation pour la Recherche en Environnement (SOERE), Les Verrines CS 80006, F-86600 Lusignan, France; marie-laure.decau@inra.fr
* Correspondence: erwan.ledeunff@unicaen.fr; Tel.: +33-2-3156-5373

**Abstract:** Under field conditions, plants need to optimize nutrient ion and water acquisition in their fluctuating environment. One of the most important variables involved in variations of ion uptake processes is temperature. It modifies the thermodynamic processes of root uptake and ion diffusion in soil throughout day–night and ontogenetic cycles. Yet, most models of nitrogen (N) uptake in plants are built from set values of microscopic kinetic parameters, $V_m$ and $K_m$, derived from a Michaelis–Menten (MM) interpretation of nutrient isotherms. An isotherm is a curve depicting the response of root nitrate influx to external nitrate concentrations at a given temperature. Models using the MM formalism are based on several implicit assumptions that do not always hold, such as homothetic behavior of the kinetic parameters between the different root biological scales, i.e., the epidermis cell, root segments, root axes, and the whole root system. However, in marine phytoplankton, it has been clearly demonstrated that the macroscopic behavior in the nutrient uptake of a colony cannot be confounded with the microscopic behavior of individual cells, due to the cell diffusion boundary layer. The same is also true around plant root segments. Improved N uptake models should either take into account the flexibility of the kinetic parameters of nitrate uptake at the cellular level (porter–diffusion approach) or use the more realistic macroscopic kinetic parameters proposed by the flow–force approach. Here we present recent solutions proposed in marine phytoplankton and plant nutrient uptake models to make a more flexible description of the nutrient ion uptake process. Use of the mechanistic porter–diffusion approach developed in marine phytoplankton introduces more flexibility in response to cell characteristics and physical processes driven by temperature (diffusion and convection). The thermodynamic flow–force interpretation of plant-based nutrient uptake isotherms introduces more flexibility in response to environmental cues and root aging. These two approaches could help solve many problems that modelers encounter in these two research areas.

**Keywords:** nitrogen; nitrate; ion transport models; logistic models; nutrient uptake-kinetics; diffusion; root growth; root aging; phytoplankton; cell populations

## 1. Introduction

Mechanistic plant modeling of root nitrate absorption in structure–function models suffers from two major biases. These are the formalism used to model nitrate transporter functioning and the

estimation of the root biomass contributing effectively to nitrate uptake. Both biases directly originate from the basic equation used to build these models. Root absorption of nitrate is usually defined by the following Equation (1):

$$\text{Net } NO_3^- \text{ influx rate} = (\text{number of } NO_3^- \text{ transporters} \times \text{transporter activity})/ \tag{1}$$
$$\text{root surface or biomass or length unit}$$

where net $NO_3^-$ influx is expressed as nitrate amount per unit time per unit surface area or biomass or length, transporter activity is defined as amount of nitrate taken up per unit time, and root surface area or biomass or length is expressed in $cm^2$ or in cm or in g. Surface areas of epidermal trichoblast and atrichoblast cells present in all root classes contribute to root surface area.

The numerator in this equation (in brackets) accounts for the functional plasticity of the model linked to changes in intrinsic activity of nitrate or ammonium transporters and their number in relation to soil nitrate concentration heterogeneity. The denominator accounts for the structural plasticity of the model related to changes in root growth. These two terms do not act on the same time scale: minute to hour for transporter activity and day to week for structural root growth plasticity.

The plasticity of the absorption thus depends as much on the spatial and temporal regulations exerted on the absorption function as on the modifications of the root structure during the development of the crop species. Commonly, root intrinsic transporter activity modeling is based on the enzyme–substrate equation (i.e., Michaelis–Menten (MM)-like kinetics with two key parameters, $V_{max}$ and $K_m$). Accordingly, this fundamental equation, mainly applied for the mechanistic construction of N uptake models, explains why values of $I_{max}$ or $V_{max}$ (maximum influx or maximum speed) and the values of root biomass or root length are the main parameters, whose sensitivities have the strongest effects on model outputs [1–5]. Models using the MM formalism are based on a first implicit assumption, not always sound, that there is homothetic behavior of the kinetic parameters between different root biological scales, i.e., the epidermis cell, root segments, root axes, and the whole root system (Figure 1). However, the kinetic parameters determined using $^{15}N$ and $^{13}N$ tracers on an entire root system of a young plant do not reflect the behavior of the different root segments and axes of that plant and cannot be reasonably transposed to older root systems [6,7]. Therefore, the up-scaling step in models of N uptake from the root segments to the entire root system requires two other implicit assumptions: (i) That the root system is an open network for nitrate uptake, and (ii) that root aging has no effect on nitrate uptake (Figure 1).

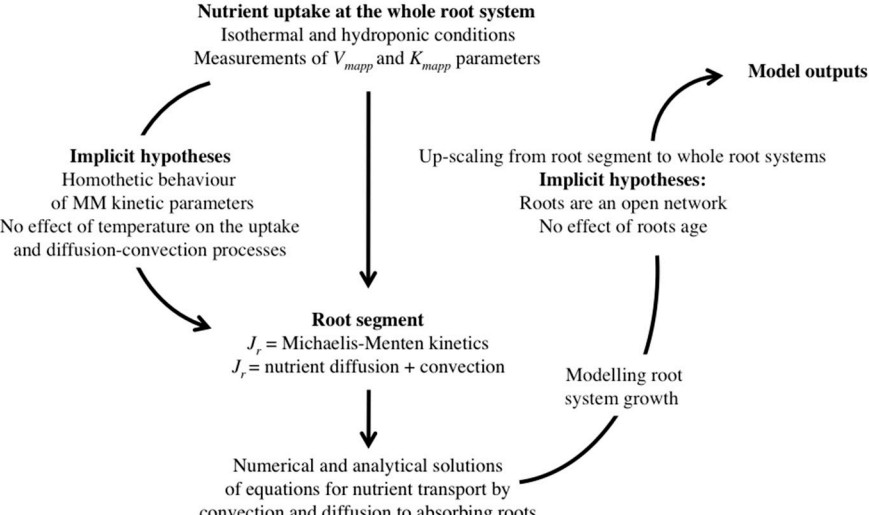

**Figure 1.** Implicit hypotheses used for building nutrient uptake models from Michaelis–Menten (MM)-type relation and convection–diffusion of nutrients toward the roots (adapted from [2–5]).

Constructing a more realistic mechanistic structure–function model requires a much-improved description of the regulation of nitrate absorption capacity and estimation (or modeling) of the root growth (or architecture) over the growth cycle. Modelers have to overcome two major pitfalls, namely overestimation of both absorption capacities at root segment level and the fraction of root segments actively involved in N absorption. Such levels of flexibility are not currently allowed for three-dimensional (3-D) models where root development is dynamically simulated through accurate subprograms such as SimRoot, RootMap, SPACSYS, R-SWMS, and RootBox [8]. Because these two variables are entangled, it is difficult to find a successful resolution unless physiologists can provide agronomists with realistic response curves of root nitrate influx according to location along the various root axes in relation to environmental changes ($NO_3^-$ concentrations, temperature, PAR, pH, etc.) and root aging. Likewise, a paradigm shift with respect to the mathematical adjustment of the spatial and temporal changes in nitrate uptake rate values in relation to the underlying processes is urgently needed [9–11].

This review presents the complementary pathways that have recently been explored for the purpose of introducing more flexibility into kinetic parameters in the modeling of nitrogen uptake in marine phytoplankton and plants. First, we show that the use of empirical models is often undervalued. The nitrate uptake kinetic parameters deduced from these models clearly show that variations in N absorption capacities during the ontogenetic cycle for a monoculture crop are related to precise phenological stages, irrespective of the level of fertilization. Thus, some phenological stages related to maximum rate of absorption and maximum deceleration can be determined from the location of leaf nodes on the main shoot axis. We go on to present two interpretations of ion uptake isotherms that can be used to deduce the kinetic parameters in standard laboratory conditions using N tracers. Here the term "kinetic" refers to nitrate root influx rate in response to a wide range of external nitrate concentrations. We show that the thermodynamic flow–force interpretation is more suitable and realistic than the Michaelis–Menten (MM) formalism for describing and interpreting nutrient uptake isotherms. We also propose two ways, that are not mutually exclusive, taken in some N uptake models in marine phytoplankton and plants to introduce more flexibility in nitrate uptake rates over long periods of time, where environmental variables such as temperature and ion concentrations fluctuate widely. One is provided by the porter–diffusion models developed in phytoplankton and embedded in the MM formalism. The other is given in plants by the cross-combination of flow–force modeling of nitrate isotherms with the effects of environmental and endogenous factors. Finally, we demonstrate that N uptake rate deteriorates with aging of roots or cells, suggesting that the whole root system cannot be considered as an "open network". The root aging effect is therefore critical in avoiding overestimated N uptake in N models during the up-scaling step from root segment to the whole root systems.

*1.1. Empirical Models Used to Determine the Kinetic Parameters of N Uptake Rate under Field Conditions for Different Levels of N Fertilization*

Under field conditions, the kinetics of N accumulation during a complete crop cycle in annual and biennial crop plants follow sigmoidal asymptotic curves (Figure 2A). These logistic, bilogistic, or sometime multilogistic functions are similar to those encountered in models of growth to describe variations in cell numbers, dimensions, biomass, or any other variables [12,13]. For example, in winter oilseed rape, N accumulation has been measured during the crop cycle on representative plant samples per $m^2$ for different levels of N fertilization under field conditions and then reported per hectare and expressed in kg N $ha^{-1}$ C·$day^{-1}$ [14,15]. Sigmoid asymptotic models of the N accumulation kinetics obtained are therefore statistical representations of the kinetic behavior of N uptake for a given genotype of a monoculture crop (Figure 2A).The processes of N absorption and accumulation directly reflect the regulations exerted on the N absorption systems (e.g., nitrate transporters) at root level in relation to changes in the environment that are similar for all N treatments, such as soil water content, nitrate concentration, soil temperature, irradiance, etc. The analysis of these statistical curves

first requires the choice of a good mathematical model of the N uptake function and an accurate estimation of the function's parameters (Table S1). In the second step, an interpretation of the chosen mathematical adjustment can be proposed in terms of temporal structure (or trajectory) of the N uptake process, which makes it possible to obtain a discretized image of the process (Figure 2D, [12,16]). Thus, the use of the pair of kinetic quantities (*v*, *g*), such as maximum absolute speed (*v*) and maximum absolute acceleration and deceleration (*g*) of N taken up enables us to describe without ambiguity the temporal structure of N accumulation [12,16]. The temporal structure of the N accumulation activity is then defined, as shown in Figure 2D, by a series of phases delimited by singular points (extrema): Maximum absolute speed (Figure 2B) and maximum absolute acceleration/deceleration (Figure 2C).

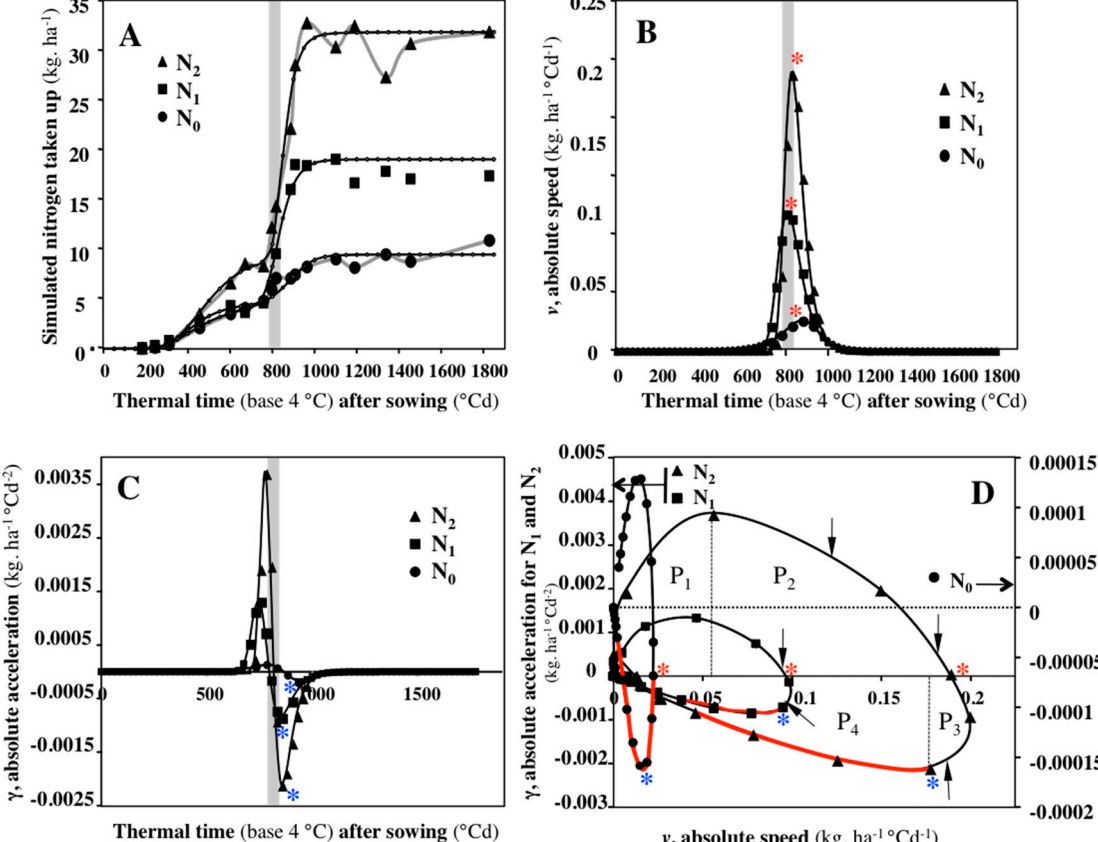

**Figure 2.** N accumulation by *Brassica napus* under field conditions after three levels of fertilization ($N_0$ = 0 kg N ha$^{-1}$; $N_1$= 135 kg N ha$^{-1}$; $N_2$ = 272 kg N ha$^{-1}$). (**A**) Specific speed of N acquisition; d$Ln$(y)/dt. (**B**) Absolute speed of N acquisition; dy/dt. (**C**) Absolute acceleration of N acquisition; d$^2$y/d$^2$t. (**D**) N acquisition trajectory (*v*, *g*). The dashed lines indicate the extrema such as maximum absolute speed (red asterisks) and acceleration/deceleration (blue asterisks) that delimit the four phases $P_1$, $P_2$, $P_3$, and $P_4$ of N acquisition during the growth cycle between the rosette stage to harvest. The grey box indicates the window of fertilization dates (787, 815, and 844 °Cd). Arrows indicate the N application dates for $N_1$ (787 °Cd = 78 kg ha$^{-1}$ and 815 °Cd = 57 kg ha$^{-1}$) and $N_2$ (45 °Cd = 49 kg ha$^{-1}$, 787 °Cd = 78 kg ha$^{-1}$, 815 °Cd = 107 kg ha$^{-1}$, and 844 °Cd = 38 kg ha$^{-1}$) fertilization treatments. The red lines indicate the N dilution in the shoot biomass deduced from Figure 3A (adapted from [15]).

## 1.2. Effects of N Fertilization on Temporal Structure of N Uptake Process in Brassica Napus Plants under Field Conditions

The determination of the kinetic quantities (*v*, $\gamma$) applied to N taken up under field conditions by a winter oilseed rape culture (cultivar 'Goëland') during the second part of the growth cycle (from rosette stage to harvest) and for three levels of fertilization N ($N_0$: 0 Kg ha$^{-1}$; $N_1$: 135 kg ha$^{-1}$; $N_2$: 272 kg

ha$^{-1}$) shows four distinct phases: P1, P2, P3, and P4 (Figure 2D). In Figure 2D, the four phases are represented only for the N$_2$ fertilization level. The results indicate that phases P1 and P4 delimited by the absolute acceleration and deceleration (Figure 2C) are predominant in duration. We note that phase P1 corresponds to the beginning of the bolding period and phase P4 is the flowering and pod filling stage. As indicated in Figure 2A–C by a grey line and arrows in Figure 2D, fertilizer is mainly applied during phase P2 (between 787 and 844 °Cd$^{-1}$). The contribution of N fertilizer does not modify the duration of the four phases of N uptake kinetics, but significantly changes the quantities of N taken up during each of these phases (Table 1).

**Table 1.** Temporal structures of N uptake during the second phase of growth (from rosette stage to harvest) of *B. napus* plants growing under field conditions after three levels of fertilization; relative importance of each phase P$_i$ phase duration and amounts of N accumulated during each phase of N acquisition. These values are calculated from data in Figure 2D.

| Phase | N Uptake Duration (in %) | | | Amount of N Uptake ( in %) | | |
|---|---|---|---|---|---|---|
| Fertilization levels | N0 0 kg N ha$^{-1}$ | N1 135 kg N ha$^{-1}$ | N2 272 kg N ha$^{-1}$ | N0 0 kg N ha$^{-1}$ | N1 135 kg N ha$^{-1}$ | N2 272 kg N ha$^{-1}$ |
| P1 | 44.0 | 41.2 | 42.6 | 22.8 | 4 | 0.7 |
| P2 | 4.1 | 2.7 | 2.7 | 26.0 | 21.8 | 23 |
| P3 | 4.1 | 2.7 | 1.4 | 29.5 | 33.4 | 22.4 |
| P4 | 47.8 | 53.3 | 53.3 | 21.8 | 40.7 | 53.9 |

In other words, the course of the morphogenetic program (defined as timing of appearance and number of organs emerging along the stem axis, for instance) remains unchanged between the N treatments. This implies that genotype × environment has no effect on the time-course of organ appearance and final number of organs. In this case, the result is a constant number of leaf nodes along the stem axis. Thus, the dates corresponding to the extrema, such as maximum absolute speed (Figure 2B) and maximum absolute acceleration and deceleration of N taken up (Figure 2C), are not adjusted temporally as a function of the level of nitrate availability (Table 1). The morphogenetic program depends mainly on the relationships between division and cell elongation at the level of root and shoot meristems. It is well known that hormones control this program during vegetative and reproductive growth and during senescence. This is exemplified in the second part of the growth cycle for a biennial crop species such as *Brassica napus* (from the rosette stage to harvest) where the date of maximum absolute rate of N uptake is not significantly modified between the N treatments (Figure 1B, N$_0$ = 875 °Cd, N$_1$ = 800 °Cd, and N$_2$ = 825 °Cd). In winter oilseed rape, this date corresponds to the middle of the bolding period, which inaugurates the beginning of elongation of the flowering ramifications and shoot senescence [17–19]. By contrast, changes in N amounts taken up during each phase (Figure 2D and Table 1) indicate that phases P2 and P4 are predominant. Phase P2 exactly corresponds to the period of fertilization for N$_1$ and N$_2$ treatments, whereas the beginning of phase P4 characterizes the progressive decrease in N uptake and corresponds to N dilution in tissue biomass (Figures 2D and 3A). At the root level, the temporal adjustment of the absorption capacities during the four phases P1, P2, P3, and P4 can occur by three types of compensation that are not mutually exclusive: (i) Local increase in the activity of nitrate carriers per unit root length, (ii) modulation of the root length actively involved in the N uptake, and (iii) structural compensation by an increase in the absorbing root length. The regulation of nitrate transporter activity in relation to N availability can only be carried out by the functional compensations (i) and (ii). Root lengths are not significantly modified between the N treatments (Figure S1) and in winter oil seed rape 60%–90% of the whole root system is set before floral stem elongation for N$_0$ and N$_2$ fertilization treatments [10].

*1.3. Effects of N Fertilization on N Dilution Curves and N Uptake Process in Brassica Napus Plants under Field Conditions*

A more common statistical approach consists in establishing, during the crop vegetative period (between the beginning of canopy closure and flowering), the N dilution curves from data for N

taken up under different levels of N fertilization by winter oil seed rape plants [20–23]. The shoot nitrogen concentration decreases with increasing shoot biomass because there is a close relationship between crop nitrogen uptake and shoot biomass accumulation rate [24,25]. Empirical and negative allometric functions have been determined for many C3 and C4 species. These relations are given by the following Equation (2):

$$\%N = aW_c^{-b} \tag{2}$$

where $W_c$ is the total shoot biomass (in t ha$^{-1}$), N is the total N concentration in the shoots (in % of Dry Weight) and parameters $a$ and $b$ drive the dilution process as the crop grows; $a$ is the plant nitrogen concentration when $W_c = 1$ t ha$^{-1}$ and $b$ is a dimensionless coefficient. Depending on the N fertilization level, the dilution curve defines three different N statuses of the crop (Figure 3A).

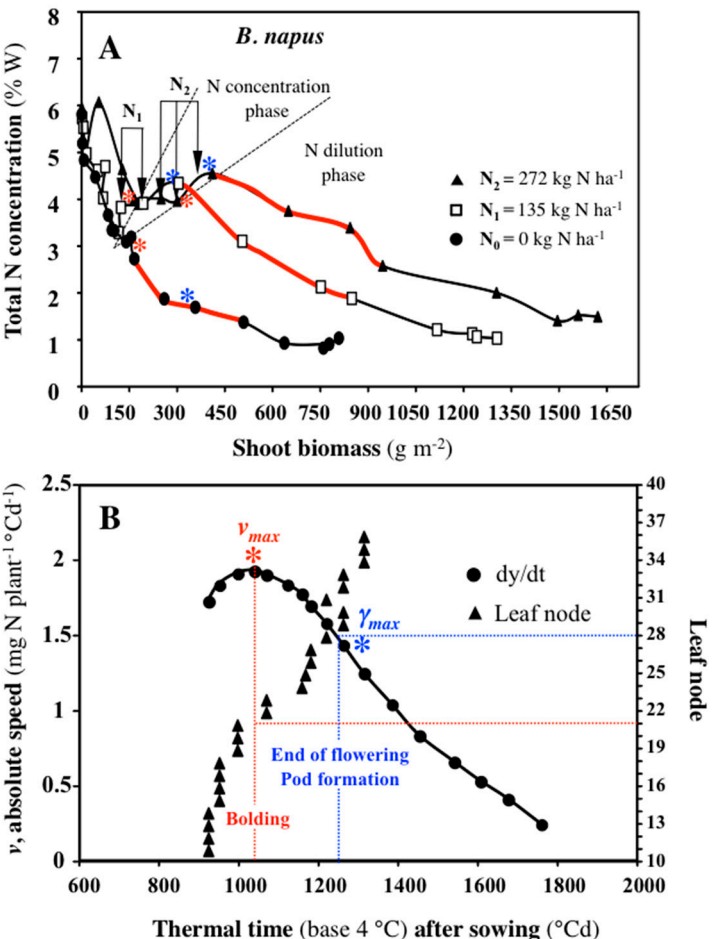

**Figure 3.** (**A**) Dilution curves of *Brassica napus* (cultivar 'Göeland') plants for three levels of fertilization (adapted from [15]). Arrows indicate the dates of fertilizer applications for N1 and N2 treatments. Red asterisks indicate the maximum absolute speed of N uptake. Blue asterisks indicate the maximum deceleration rate for each nitrogen fertilizer treatment. Red lines represent the N dilution phase from the bolting period to flowering. (**B**) Relationship between the maximum absolute speed ($v_{max}$ and $\gamma_{max}$) of the N uptake rate and the beginning of N remobilization (date in thermal time) of a specific leaf node in *Brassica napus* (cultivar 'Capitol', adapted from [26]).

In Figure 3A, in the top curve, the shoot growth is not limited by N (N$_2$ = 272 kg ha$^{-1}$), in the middle curve, the N concentration is close to the optimum (N$_1$ = 135 kg ha$^{-1}$), and in the bottom curve, growth is limited by N supply (N$_0$ = 0 kg ha$^{-1}$). Again, we emphasize that the same type of conclusion can also be drawn from the parallelism of dilution curves for the three levels of fertilization. Dilution curves clearly demonstrate, like for N uptake trajectories (Figure 2D), that the course of the

shoot morphogenetic process (i.e., the number of leaf nodes along the stem axis) is not affected by the different N treatments (Table 1). We note that the dilution curves could not explain the nitrogen concentration phases in shoot tissues induced by the different levels of nitrogen fertilization (Figure 3A). In particular, unlike the temporal structure approach (Figure 2D), this type of approach can provide no explanation on the phenological stages when the maximum absolute speed and deceleration of N uptake are reached during the N concentration phase of shoot tissues. However, these extrema related to the phenological growth of the main stem and flowering ramifications explain the output levels of N dilution curves in relation to N fertilization levels (asterisks in Figure 3A). Adjustments between the temporal structure of N uptake and nitrogen dilution curves for levels of fertilization $N_1$ and $N_2$ reveal that the nitrogen dilution starts exactly when absolute deceleration of N uptake occurs (compare Figures 2D and 3A). Furthermore, on another set of field data in *B. napus* (cultivar 'Capitol'), the maximum N uptake rate ($\nu_m$ in Figure 3B) is reached when the leaf at node 21 on the main stem starts to export its accumulated nitrogen (Figure 3B). As previously reported [26], the contribution of nitrogen provided by root uptake to leaf N accumulation begins to decrease from leaf nodes 21–22. Nitrogen accumulated in leaf nodes 22–36 from older leaves is exclusively involved in flowering and pod filling. Leaves 22–36 also exhibit the highest values of N remobilization, and leaf nodes 26–28 ($\gamma_{max}$ in Figure 3B) inaugurate the progressive decrease in the duration of remobilization [26].

Based on the dilution curves, a theoretical relationship between crop N uptake and shoot biomass has been established, as these two processes are co-regulated by soil N availability and crop growth capacity [22]. Using the Michaelis–Menten equation to formalize the N uptake process in response to N availability and by derivation of Equation (2), it is possible to obtain the following Equation (3):

$$dNupt/dt = (a\,(1-b)\,W_c^{-b}) \times ((dW)_{max}/dt) \times [V \times C]/(K + C) \tag{3}$$

where the first term of the equation represents the maximum biomass growth rate and the second term depends only on N absorption regulated by the excess of N availability. Using reasonable simplifying assumptions such as (i) root length density does not limit nitrate uptake, (ii) nitrate is evenly distributed across the different soil layers, and (iii) moisture of soil is not limiting, it is possible to deduce the values of $V_m$ and $K_m$ parameters of the second term of Equation 3. In most cases used for estimations [22], these simplifying assumptions were only fulfilled for the first layer of soil (0–30 cm). There was greater heterogeneity for these variables in the deeper soil layers and most roots were localized in the top layers [22,27,28], except in the absence of N fertilization where the environments in the different soil layers probably became more homogenized. This approach, used for winter oil seed rape growing under field conditions, allows patterns of N uptake rate index (NUI) to be determined for a wide range of nitrate concentrations in the first soil layer [22]. Although the response of NUI to nitrate concentration was well described by a single hyperbolic function (monophasic model 1) and a double hyperbolic function (biphasic model 2), a logarithmic function gave a better fit for the experimental data points (model 3, Figure 4A). This is best illustrated using semi-logarithmic coordinates {Nupt/dt; *Ln* $[NO_3^-]_{ext}$} (Figure 4B), and suggests that ion uptake by roots can be better described using a *flow–force* type law based on thermodynamics [29] (see § 3.2).

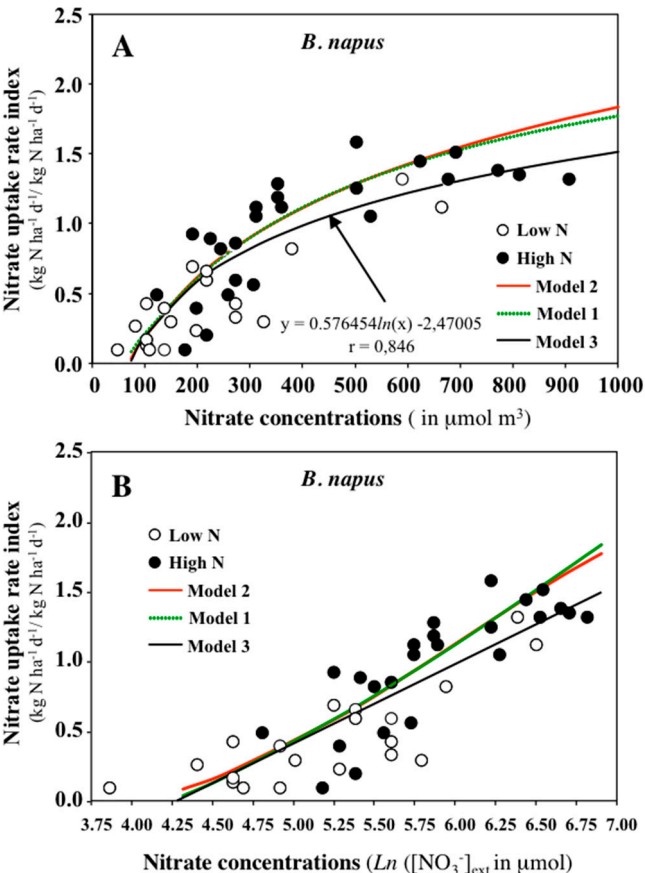

**Figure 4.** (**A**) Relationships observed between nitrate uptake rate index (NUI) and soil nitrate concentrations in 0–30 cm soil layer for oilseed rape (*B. napus*) in field experiments (adapted from [22]). The different models correspond to mathematical adjustments with a single hyperbolic function (model 1), a double hyperbolic function (model 2), and a logarithmic function (model 3). (**B**) Linear transformation in semi-logarithmic coordinates $\{N_{upt}/dt; [NO_3^-]_{ext}\}$ of the relationships obtained in A. The linear curve in model 3 is based on the flow–force interpretation of ion transport in plants (adapted from [29]). In this interpretation, the slope of the linear curve in model 3 is a measurement of the overall conductance ($L_j$) of the root system for nitrate and the intersection of plot on the abscissa (i.e., $N_{upt}/dt = 0$) enables us to estimate a thermodynamic parameter $\pi_j$ (adapted from [11,29]; see § 3.2).

The slope break observed with a double hyperbolic function (model 2, Figure 4A) at around 0.5–1 mM of external nitrate concentration is commonly interpreted as a dual enzymatic mechanism of ion uptake [30]. However, this can also be interpreted as the transition from diffusion to the convective regime during the mobility of nitrate from soil to roots (Figure 4A). Given the effect of diffusion in phytoplankton ion uptake (see below, [31]) and the effect of nitrate in aquaporin activity and plant water uptake [32–34], it is likely that diffusion and convection processes have been underestimated in explaining biphasic pattern of ion uptake rate in plants. Nitrate in the soil passes through a boundary layer around the root in which nitrate transport towards the root depends mainly on diffusion. The thickness of the diffusion boundary layer depends on active nitrate uptake at the root plasma membrane, but also on the soil nitrate mass flow (convection) driven by the transpiration stream. High convective nitrate flux increases the nitrate concentration towards the root and diminishes the diffusive boundary layer [2,3].

In summary, the statistical approaches to N uptake kinetics for different levels of N fertilization indicate that the morphogenetic program is rigid under given agro-climatic conditions (genotype × environment interactions) for a monoculture crop. The morphogenetic program induces a functional spatial and temporal regulation of N uptake to adjust the quantities of N taken up during each phase of

the N uptake trajectory (temporal structure) or shoot growth (N dilution curve) to the soil N availability, as shown in empirical models. Accordingly, the old formula, "the N uptake adjusts to the N demand" is wrong if we do not consider kinetic processes as a whole with adapted descriptors ($v$, $\gamma$) in the comparisons between N treatments and by adjusting the evolution of these descriptors with accurate phenological stages. Based on the kinetic analyses (Figure 2) and root biomass data (Figure S1), we see that the N uptake adjusts to the N supply (provided by mineralization and fertilization) without changes in the morphogenetic program. In *B. napus* plants, nitrate uptake rate depends mainly on soil nitrate availability and on the functional, rather than structural, compensations of N uptake capacities.

### 1.4. Origins of the Overestimation of N Uptake Capacities in Most Mechanistic Models

In soils where the nitrate concentration is not limiting and the water content is sufficiently high, calculations of the inflow rates of nitrate based on N supply and N demand in relation to the length of the root system are most often overestimated [35,36]. The use of total root length in calculations strongly overestimates the active root fraction actually involved in N uptake, which leads to an underestimation of the actual inflow rate in the truly active root zones [36]. Theoretical estimates of inflow rates can be obtained by incrementally reducing the fraction of the active roots involved in N uptake to match the root lengths with the accumulated total N amounts. In this type of approach, 11% and 3% of total root length in wheat (*Triticum aestivum* L. cultivar 'Wembley') have been estimated to participate in absorption from unfertilized ($N^-$) and fertilized soils with 200 kg N ha$^{-1}$ ($N^+$) [36].

Similarly, in the 1-D N-absorption models, the absence of regulations on the space-time dynamics of the absorption of the root system explains why, during sensitivity analyses, the root biomass and values of $I_{max}$ and $V_{max}$ (maximum velocity) are the two parameters whose sensitivity is the highest on model outputs [2,3]. These two types of approach lead to circular reasoning, since it is very difficult to establish experimentally whether the adjustment of inflow rates should be done on the active root fraction or nitrate absorption rate ($V_{max}$ value). How these two variables vary in space and time relative to each other in response to N resource availability remains one major unresolved question that plant physiologists have still to address. Experimentally, it has been shown that the inflow rates are not identical along the root axes [37–40]. There may be several physiological reasons for these differences, such as epidermal cell suberization, cortical cell senescence, and some root surface fractions that are not in close contact with the soil solution [36]. Hence the variations in $V_{max}$ values and determination of those root fractions that are really active in the absorption of N are the two critical points of all N absorption models currently developed.

### 1.5. How Do We Interpret the Kinetic Parameters Deduced from the Mathematical Adjustment of Ion Uptake Isotherms?

In the case of nitrate, the kinetic parameters of the absorption can be measured in the laboratory on the roots using stable ($^{15}N$) or radioactive ($^{13}N$) isotopes in hydroponic solutions under standard conditions. In general, measurements are made on the root system of young plants in isothermal conditions for a given light intensity and a specific time during the day–night cycle. Thus, for different external nitrate concentrations, different values of nitrate influx rate are obtained. The experimental values are then adjusted using mathematical curves and the kinetic parameters are interpreted. These curves are called ion uptake isotherms. Two major interpretations have so far been proposed: The enzyme–substrate interpretation based on Michaelis–Menten (MM) enzyme kinetics [41–43] and the flow–force interpretation based on thermodynamics of ion transport across the roots [29,44–48]. Even though these two interpretations describe the isotherms using only two parameters, they differ completely in essence. The enzyme–substrate interpretation of nutrient uptake isotherms seeks to infer microscopic functioning of ion transporters at the root epidermis, whereas the flow–force interpretation describes macroscopically the functioning of the whole root catalytic device, without inferring any specific microscopic behavior. Similarly, analogical reasoning with the enzyme–substrate interpretation of ion uptake isotherms with the MM equation in microorganisms such as marine phytoplankton

has also led to a wrong interpretation and misappropriation of the phenomenological model of bacterial growth [49–51].

*1.6. Enzyme–Substrate Interpretation of Ion Uptake Isotherm: Deduction of Microscopic Parameters Vm and Km*

The mathematical adjustment with square hyperbolas of the points obtained from experimental inflow rate measurements at different external nutrient concentrations led Epstein and coworkers to interpret ion transport as a dual enzymatic mechanism [30,41,52,53]. A first enzymatic mechanism observed between 0 and 1 mM external nitrate concentration defines a high affinity transport system (HATS) and a second enzymatic mechanism observed between 1 and 20–30 mM external nutrients defines a low affinity transport system (LATS) based on analogical reasoning with the enzymatic formalism of MM:

$$S_j^e + C \underset{k_{-1}}{\overset{k_1 \quad \rightarrow}{\leftrightarrows}} \quad CS \quad \overset{k_2}{\rightarrow} \quad C + S_j^i \tag{4}$$

where $C$ is the carrier, $S_j^e$ the substrate outside the cell, $S_j^i$ the substrate incorporated into the cell by the carrier, and $k_1$, $k_2$, and $k_3$ are reaction velocity constants. As part of this enzyme–substrate interpretation, microscopic kinetic parameters such as $V_{max}$ and $K_m$ for nutrient absorption transport systems can be deduced according to:

$$V = k_2 C [S] / ((k_{-1} + k_2)/k_1 + [S]) = J_j^{ei} (c_j^e) = V_{max} [c_j^e] / (K_m + [c_j^e]) \tag{5}$$

where $V_{max} = k_2[S]$ is the saturation velocity of carrier C by substrate $S_j$, $c_j^e$ is the external concentration for substrate $S_j$, and $K_m = (k_{-1} + k_2)/k_1$ is the ratio of reaction velocity constants and defines the affinity of carrier C for substrate $S_j$ (half-saturation constant).

Although the behavior of an entire root system or cell population is difficult to assimilate to a purified enzyme in a test tube in the presence of an excess of substrate as in enzymatic experiments, the enzyme–substrate interpretation of ion uptake isotherm kinetics with MM formalism has prevailed among plant and phytoplankton physiologists for more than 50 years [30,43]. Notably in plants, values of $K_m$ and $V_{max}$ are always used to model ion uptake [2–5,54,55] as in ecological models of marine phytoplankton [42,43,56–60]. However, in marine phytoplankton it has been shown that values of kinetic parameters estimated from isotherms are extremely variable even for the same species and there is a co-variation between $V_{max}$ and $K_m$ Therefore, the affinity of a species for a nutrient is now better characterized by the slope $a = V_{max}/K_m$ (i.e., the ratio of constant $V_{max}/(k_{-1} + k_2)/k_1$) at the origin of the graph, rather than $K_m = (k_{-1} + k_2)/k_1$ because $\alpha$ is a better approximation of $1/Km$ ($k_1/(k_{-1} + k_2)$). This new affinity constant is interpreted as the sum of the total affinity of all carrier sites present at the cell surface and is a better indicator of nutrient uptake efficiency [56,57,61–64]. Similarly in plants, the use of the parameter $K_m$ (i.e., $K_m = (k_{-1} + k_2)/k_1$) to characterize the affinity of the roots for a nutrient is very questionable, since $K_m$ values vary over one order of magnitude within the same species between different experiments (Table 2).

**Table 2.** Michaelis–Menten half-saturation constants and maximum velocity for nitrate uptake from publications in barley (*Hordeum vulgare*) and *Arabidopsis thaliana*.

| References | NO$_3^-$ Uninduced Plants | | NO$_3^-$ Induced plants (1 mM 6–24 h) | |
| --- | --- | --- | --- | --- |
| | $K_m$ (µmol) | $V_{max}$ (µmol h$^{-1}$ (DW or FW) g$^{-1}$) | $K_m$ (µmol) | $V_{max}$ (µmol h$^{-1}$ (Dry or Fresh Weight)g$^{-1}$) |
| *Hordeum vulgare* | | | | |
| [65] | nd | nd | 7–62–86 * | 0.5–1.5–2.5 *(DW) |
| [66] | 6.4–7.8 | 21.7–22.4 (FW) | 12–14 | 43.9–48.7 (FW) |
| [67] | 20 | 0.34 (FW) | 79 | 9.4 (FW) |
| [68] | 6 | 0.82 (FW) | 36 | 8.4 (FW) |
| *Arabidopsis thaliana* | | | | |
| [69] | nd | nd | 92 | 1 (FW) |
| [70] | nd | nd | 52.2 | 147 (DW) |
| [71] | nd | nd | 14.1 | 11.16 (FW) § |

* The following values correspond to three different cultivars (Olli, Kombar, and Prato). Nd—not determined.
§ Plants were induced with 1 mM KNO$_3$ for 6 hours.

## 2. Flow–Force Interpretation of Kinetic Parameters of Nutrient Isotherms: Deduction of L$_j$ and $\pi_j$ Macroscopic Parameters

In the 1970s, an alternative approach was proposed to model the ion uptake kinetics [29,44,48]. The *flow–force* approach is based on a thermodynamic interpretation of the absorption, rather than an enzymatic process. Using reasonable simplification assumptions, the flow of a substance can be written as the following equation (for details see [29]):

$$J_j(c^e_j) = RT\lambda_j \ln((c^e_j)/(^\circ c^e_j)) = L_j \ln((c^e_j)/(^\circ c^e_j)) \tag{6}$$

With

$$L_j = RT\lambda_j \tag{7}$$

where $R$ is the gas constant, $T$ the absolute temperature, and $\lambda_j$ is the overall conductance of the sample for the net uptake of $S_j$. This means that when a system of semi-log coordinates is used {ln($c^e_j$), $J_j(c^e_j)$}, the plot representing the experimental points of the ion uptake isotherm is expected to be quasi-linear for the values of $c^e_j$ sufficiently close to the equilibrium concentrations $\circ c^e_j$ [11,29]. The flow of nutrients through the membrane is therefore defined by two macroscopic parameters: The overall conductance of the root system for the substrate ($L_j$) and the thermodynamic parameter ($\pi_j$) when $J_j$ = 0. The parameter $\pi_j$ equal to $1/c^e_j$ represents the contribution of all terms other than $c^e_j$ involved in the driving forces energizing the absorption of $S_j$ if the equilibrium is not completely respected. Accordingly, the flow–force interpretation infers neither the hypothetical cellular processes within the root cell layers nor the type of carriers involved in nutrient transport, but proposes a macroscopic interpretation (phenomenological model) of root uptake behavior based on a thermodynamic approach.

The kinetic parameters of nutrient uptake isotherms in enzyme–substrate and *flow–force* interpretations are determined from the measurements of inflow rates for the whole root system. A major question therefore arises of how kinetic parameters scale with different levels of root organization in response to external nitrate concentrations and environmental factors. There is no particular reason why the values of kinetic parameters should exhibit homothetic behavior across the different biological scales used for influx rate measurements.

## 3. Meaning and Validity of Kinetic Parameters of Nutrient Isotherms at Any Root Scale

In plants, as illustrated in Figure 5A, the kinetic parameters of the enzyme–substrate interpretation obtained for the whole root system are in fact macroscopic parameters (apparent $V_m$ and $K_m = V_{mapp}$

and $K_{mapp}$) because they represent subsumed activities of multiple transporter types in the plasma membrane of root epidermis cells present along the root axes [11,72].

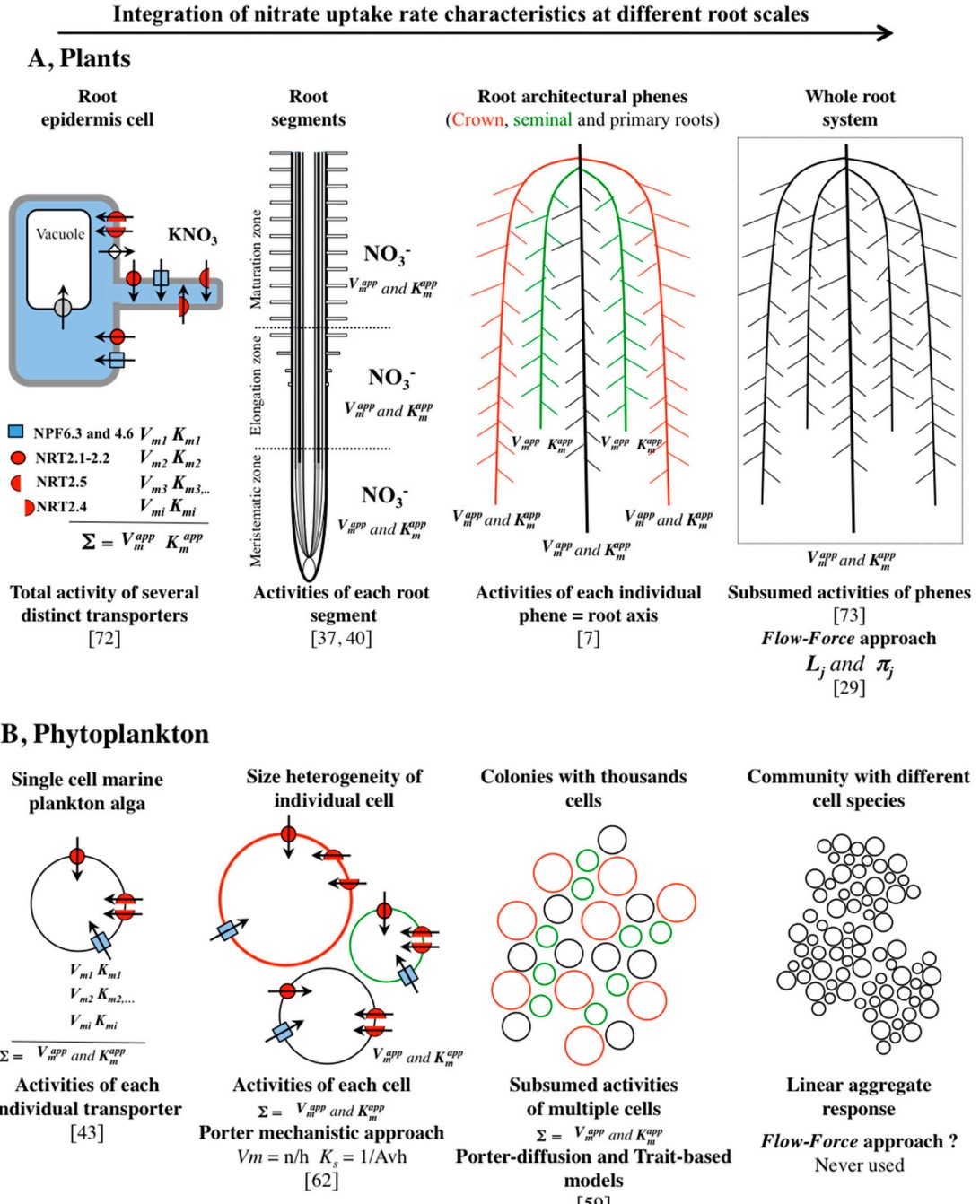

**Figure 5.** Integration of kinetic parameters for the nitrate uptake rate at different biological scales. (A) From the root epidermis to the whole root system in plants. (B) From individual cell to colony and community in marine phytoplankton. In the MM equation: $V_{mapp}$, is the apparent maximum velocity, $K_{mapp}$ and $K_s$ are the apparent half-saturation constants. In the porter–diffusion equation: $A$ is the area of one uptake site, $n$ is the number of ion uptake sites per cell, $v$ is the mass transfer coefficient, $h$ is the handling time per ion on an uptake site.

Accordingly, the parameters of N uptake rate, estimated with isotopic tracers, are not microscopic parameters that would be valid for all experienced root scales (e.g., from the epidermis to root segments and from the root segments to the whole root system). In other words, they do not have the meaning

we might expect ($V_{max}$ and $K_m$ from an enzymatic reaction due to one single transporter). Even at the root epidermis (Figure 5A), the N influx rate of a single cell is the subsumed activities of different types of transporter arranged in series at the plasma membrane ([72], see Chap. 6; Equation 107, 112). For example, recent molecular studies show that there are at least four different gene transporter families (*NPF*: Nitrate transporter1/peptide transporter family; *NNP*: Nitrate/nitrite porter family; *CLC*: Chloride channels; *SLAC1/SLAH*: Slow anion channels/homolog, and *NAX*T: Nitrate excretion transporter) involved in nitrate uptake. The genes encoding these transporters are redundant and operate, e.g., *NRT2.1, NRT2.2, NRT2.4, NRT2.5, NRT1.1* (*NPF6.3*) and *NRT1.3* (*NPF6.8*) in low and high ranges of external nitrate concentrations [71,73–77]. Moreover, the activities of these transporters can be coupled with each other or not. Depending on the biological scale used in experiments, the values of nitrate uptake parameters are therefore not fixed but vary continuously. This is well illustrated by changes in the inflow rates between root segments along the primary root [7,37,40]. We are thus unable (i) to find a simple meaning for these MM parameters in relation to the integrated constitution and functioning of the root sample at a molecular level and (ii) to fill the gap between the individual activity of each type of transporter (real microscopic parameters) at the root epidermis level and the influx measured at higher biological scales (macroscopic parameters). It is also likely that the physics of the system, such as diffusion, convection, and temperature, dominate the behavior of the kinetic parameters such as $V_{mapp}$ in soil, as observed for marine microorganisms [31,78].

In MM models of marine phytoplankton, the same pitfalls related to the biological scale used are encountered in characterizing the parameters of ion uptake kinetics (Figure 5B). There is no way to determine how the values of kinetic parameters scale with different levels of organisation: Single cell, colonies, or community of phytoplankton species, in response to wide ranges of N external concentrations [58,59,62,79,80]. For nearly 50 years, the MM model introduced by Dugdale [43] in phytoplankton prevailed as the standard formalism for the modeling of growth and nutrient uptake. This stems from an initial confusion between the MM model of enzymatic reaction and the phenomenological model of bacterial growth [51] to simplify and approximate the model of Teissier [49,63]. Despite the warnings of several authors [61,63,64,81], this confusion persisted. This is why the use of MM models was challenged as the best functional representation of N uptake at various biological scales [61,62,79,80]. MM models used to describe nutrient uptake behavior measured in bulk external nutrient concentration ignore the boundary layer of phytoplankton cells, where ion diffusion properties, cell size, and temperature play major roles [31,58,59,82]. Uptake rate can therefore be better characterized by a quadratic equation than an MM equation (see § 4). Furthermore, at the molecular level, phytoplankton cells have the same families of nitrate transporters [83–86] as those found in plants such as NNP (nitrate/nitrite porter family) and NPF (nitrate transporter1/peptide transporter family). The nitrate uptake parameters thus correspond to subsumed activities of different nitrate transporters arranged in series on the phytoplankton plasma membrane. At the cell colony and community levels, $V_{max}$ and $K_m$ must also be considered as macroscopic parameters ($V_{mapp}$ and $K_{mapp}$) because they represent resultant activity of individual uptake of the colony cells or community multi-specific cells (Figure 5B). New trait-based models were accordingly built where nutrient uptake is explicitly parameterized in terms of cell size, uptake sites, and molecular ion diffusion (see below § 4.).

Because many phenomena are commonly interpreted in physics (e.g., Ohm's law, Darcy's law, etc.) and in biophysics (e.g., Nernst–Planck equation, water transport in plants, etc.) by general *flow–force* laws, the thermodynamic interpretation of ion transport proposed by Thellier [44,48] is a more realistic alternative to describe the macroscopic behavior of the N uptake rate irrespective of biological scale. Its linear formalism is moreover simpler for building N uptake models on a macroscopic scale where physical parameters play a major role. Unlike MM formalism, *flow–force* parameterization through nutrient conductance ($L_j$) across cell membranes embeds the temperature as an important state variable (see Equation (6)).

However, it must be underlined that whichever interpretation is chosen for the nutrient uptake isotherms (flow–force versus enzyme–substrate) and the biological scale used, the static nature of the kinetic parameters ($L_j$ and $\pi_j$ versus $K_{mapp}$ and $V_{mapp}$) for a given temperature leads to under- or over-estimations of spatial and temporal absorption capacities in the nutrient uptake models of plants and phytoplankton in response to endogenous and environmental factors. For example, in winter oilseed rape (*B. napus*) the thermal amplitude during a crop cycle is more than 30 °C (Figure S2). The next question is therefore how we can introduce spatial and temporal flexibility in the ion uptake process.

## 4. Introduction of Spatial and Temporal Flexibility in ion Uptake Rate Modelling

In this section, we explain how models in phytoplankton and plants have dealt with the rigidly set values of the macroscopic kinetic parameters to introduce a flexible uptake process in response to the dynamic spatial and temporal changes in nitrate concentration and/or endogenous and environmental factors.

### 4.1. Introduction of a Flexible Uptake Process by Regulating the Number of Uptake Sites at the Cell Membrane Level in Response to Changes in Nutrient Concentrations

In marine phytoplankton, some ways to overcome the lack of flexibility of $V_{mapp}$ and $K_{mapp}$ kinetic parameters expressed have been found in more plastic MM models [58,59,62,87]. These new models overcome the constant set values of MM kinetic parameters and allow a better plasticity of nutrient uptake in widely fluctuating environments such as oligotrophic and eutrophic regimens [58,59,87]. Lack of flexibility in ion uptake processes has been tackled by taking into account the ion diffusion process, cell size, and regulation of the number of uptake sites present in the plasma membrane. The boundary layer that surrounds the phytoplankton cells modifies the ion concentration at the cell surface $[S_0]$ from the bulk concentration $[S_\infty]$ (nutrient concentration outside the boundary layer of the cell); the number of uptake sites modulates the ion flux within cells when the diffusion process is limited or not. In other words, these new models introduce additional parameters into the absorption rate equation, but the determination of these parameters is less easy.

### 4.1.1. Plastic MM Model Formalism

Bonachela et al. [87] obtained an approximate-solution MM model that is valid both for the oligotrophic (diffusion limitation) and eutrophic regimens (saturation of carrier activity at the plasma membrane or uptake site limitation). In their plastic MM model, nutrient uptake rate is expressed as:

$$V = V_{max} [S] / (K_S (1 + V_{max}/4\pi r_0 D K_S) + [S]) = V_{max} [S] / (K_S + [S]) \tag{8}$$

where $V$ is the substrate uptake rate (in mol cell$^{-1}$ s$^{-1}$), $[S]$ represents the external nutrient concentration, $V_{max}$ is the maximum uptake rate, $K_S$ is the half saturation constant, $D$ is the diffusion constant of the nutrient (in m$^2$ s$^{-1}$), and $r_0$ is the cell radius (in m). This plastic MM formalism allowed the behavior of $V$ and $V_{max}$ to be determined for different external nutrient concentrations $[S_\infty]$ (see Figure 2 in [87]). The simulation shows that $V_{max}$ is not constant, but varies with $[S_\infty]$. Thus, for small nutrient concentrations (oligotrophic regimen), $V_{max}$ increases because the ion uptake is limited by nutrient diffusion ($D$ or $[S_\infty]$ very small). Equation 4 takes the simplified form:

$$V = 4\pi r_c D [S_\infty] \tag{9}$$

In this situation, the number of nutrient carrier sites increases to reach a maximum that is limited by the cell surface area. For large nutrient concentrations (eutrophic regimen), $V_{max}$ reaches its lowest value and $K_S$ becomes very small compared to $[S_\infty]$. In this case, the number of carrier sites only limits the ion uptake.

The main problem with this type of MM model is that it takes no account of the effects of major environmental factors such as temperature and light intensity and inherent traits such as organism size [58,59,82,88,89]. Temperature affects $V_{max}$ through the ion diffusion and metabolism processes, and light intensity modifies photosynthesis, which favours ATP production via $H^+$-ATPase. $H^+$-ATPase energizes the ion transport, which ultimately also impacts on $V_{max}$. Experimentally, it has been shown that $K_S$ values depend on the temperature and diffusion [31] and vary by more than two orders of magnitude between experiments within similar species (see review [80]).

### 4.1.2. Trait-Based Model Formalism

To correct static parameters of MM models, non-MM models were built from a mechanistic approach of N uptake functioning based on Holling's "disk" equation at the cellular level [62]. These models, also called porter–diffusion models, were built with new parameters derived from inherent phytoplankton cellular traits such as the number of uptake sites at the cell surface ($n$), the uptake site handling time ($h$), the uptake site radius ($s$), the cell radius ($r_0$), and the mass transfer coefficient ($v$) [58,62,79,90,91].

To take these porter–diffusion models one step further, the porter–diffusion model was combined with MM formalism to create trait-based models that respond to oligotrophic and eutrophic nutrient regimens [59]. The model gives a quadratic expression of ion uptake rate that can be approximated by an MM model. This approach established fundamental scaling relations in enzyme kinetic interpretation of MM models between the cellular and colony levels. In this trait-based model, nutrient uptake rate is expressed as:

$$V = V_{max} [S_\infty]/(K_0 + V_{max} (4\pi r_0 D^{-1} + [S_\infty])) \tag{10}$$

where $V$ is the substrate uptake rate (in mol cell$^{-1}$ s$^{-1}$), $K_0$ is the MM half-saturation coefficient with $S_0$ as the reference concentration (in mol m$^{-3}$), $r_0$ is the cell radius (in m), and $D$ the diffusivity of the substrate (m$^2$ s$^{-1}$).

In this equation, $V_{max} = n^{-1}$, as in the model of Aksnes and Egge [62], and $K_\infty = K_0 + n^{-1}(4\pi r_0 D)^{-1}$, where $K_\infty$ is the MM half-saturation coefficient $K_0$ for $[S_0]$ extended to the observed half-saturation coefficient $K_\infty$ for $[S_\infty]$. In regard to the plastic MM model formalism proposed by Bonachela et al., [92], this trait-based model could be used to establish, in oligotrophic and eutrophic regimens, how the relationship between the number of uptake sites and the cell size varies (see [59] for details). From the different equations provided by the model ($V_{max}$, $K_\infty$, and $\alpha_\infty$), it is also theoretically possible to estimate the molecular parameters such as $n$, $s$, and $h$ at the cell level and to introduce state variables such as temperature (see [59] for details).

### 4.2. Cross-Combination of the Flow–Force Theory with Temporal Variations of Root N Uptake Rate in Response to Environmental Changes

In plants, it was experimentally demonstrated with $^{15}N$ and $^{13}N$ tracers that the values of parameters $V_{maxapp}$ and $K_{mapp}$ are not fixed and static, but also fluctuate widely during the growth cycle depending on the environmental conditions, such as variations in day–night and ontogenetic cycles [93–95], the spatial and temporal availability of nitrate [67], plant nitrogen status [73], and variations in light irradiance and temperature [9,95]. To take into account the dynamic and flexible nature of N uptake processes during the plant growth cycle, response curves of nitrate inflow rate to changes in temperature, photosynthetically active radiations (PAR) (exogenous factors), and day-night and ontogenetic cycles (in planta regulations) were formalized using polynomial equations (Figure 6).

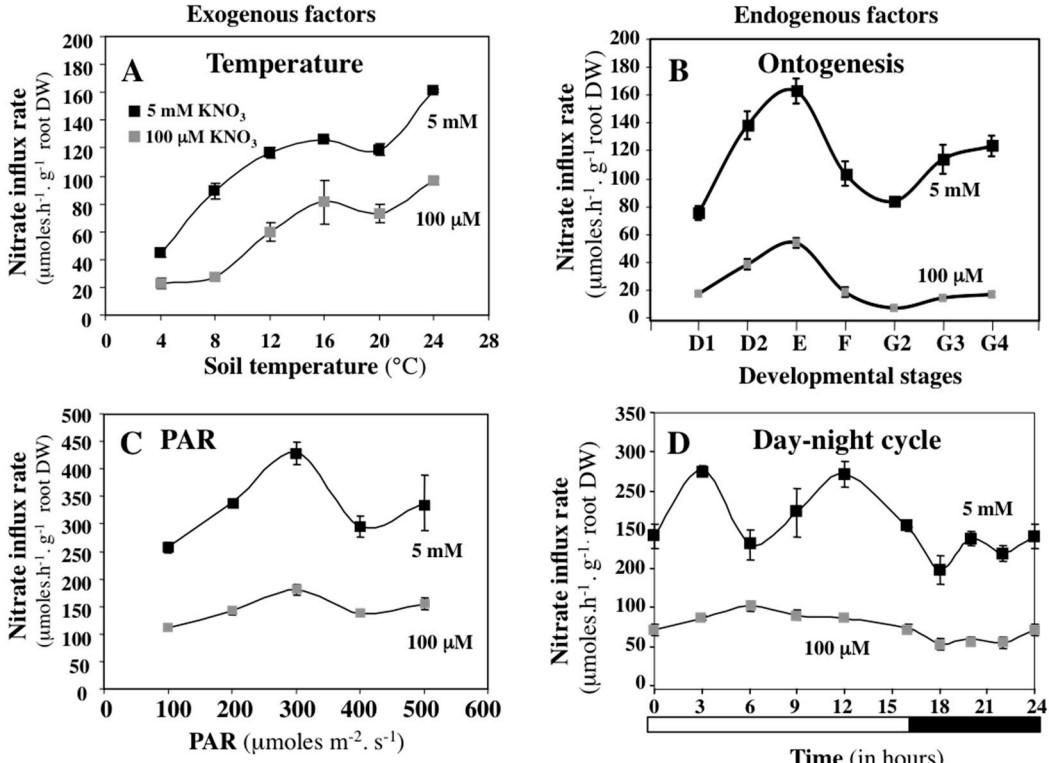

**Figure 6.** Effects of changes in exogenous and endogenous factors on $^{15}NO_3^-$ influx rates of *B. napus* plants at 100 μM and 5 mM external nitrate concentrations. (**A**) Effects of temperature. (**B**) Effects of the developmental stage during the ontogenetic cycle. D1 to D2 inflorescence formation, F1 to G2 flowering and G3 to G4 pod formation according to the phenological calendar established by the Bayer, BASF, Ciba, and Hoechst companies. (**C**) Effects of the photosynthetically active radiation (PAR). (**D**) Effects of day–night cycle. Vertical bars indicate ± SD for *N* = 3 repetitions when larger than the symbols (adapted from [19]).

Flow–force modeling of nutrient uptake isotherms was cross-combined with environmental and in planta regulation of nitrate uptake rate as depicted in Figure 7. It allows the construction of a model based on three-dimensional (3-D) influx response curves corresponding to most environmental conditions (temperature, PAR, and $[NO_3^-]_{soil}$) encountered by plants at different external soil nitrate concentrations under field conditions [9,10,19]. This modeling approach introduces a greater flexibility in the N uptake process in response to fluctuation of soil nitrate concentrations and endogenous and environmental factors [11].

Integration of the day–night regulation of nitrate influx variations allows a temporal scaling of the N uptake rate from hour to day as a function of external nitrate concentrations [9,19]. This daily time step could then be extended to the entire growth cycle by taking into account ontogenetic variations in nitrate uptake rate related to the plant morphogenetic program. The temporal variations of N uptake rate during day–night and ontogenetic cycles depend on pleiotropic effects associated with the translocation of N and C assimilates between the shoots and roots for growth, the long-distance transport of signaling molecules associated with N status, and changes in the root energy status (sugar availability).

In summary, in phytoplankton, a porter–diffusion sub-model introduces more flexibility into nitrate uptake processes at the cellular level. This sub-model uses a mechanistic approach where the cell reacts to changes in external ion concentrations induced by the processes of diffusion–advection, and regulates the number of ion uptake sites and their properties. This mechanistic sub-model is then embedded (up-scaling) in an MM formalism to create a trait-based model. In plants, flexibility in root

nitrate uptake rate in N models is obtained by mathematical cross-combinations between flow–force interpretation of nitrate isotherms (with macroscopic parameters $L_j$ and $\pi_j$ and a linear formalism) with polynomial response curves of nitrate influx to different environmental and in planta factors. The next step with these models will be to combine their respective approaches to create N uptake models that will be flexible at different biological scales.

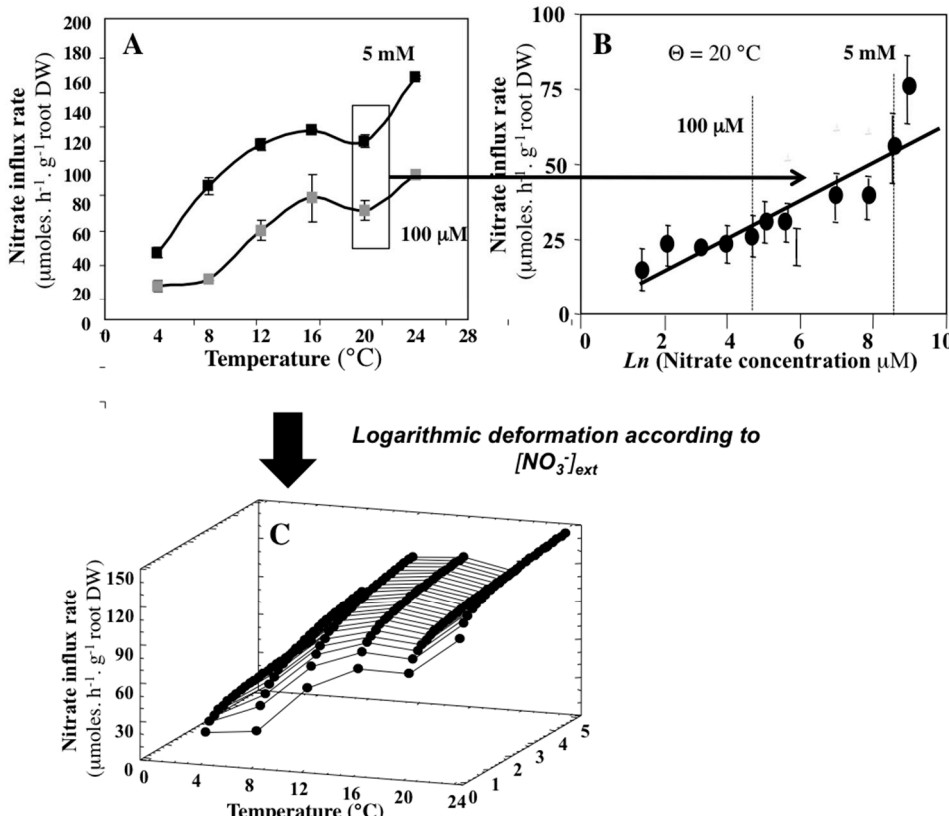

**Figure 7.** Example of a cross-combination between temperature effect on nitrate influx rate and external nitrate concentrations within a flow–force framework. (**A**) Temperature effect on nitrate influx rate at 100 μM and 5 mM. (**B**) Nitrate influx rate isotherm with *B. napus* plants at B4 stage, grown at 20 °C and 300 μmol m$^2$ s$^{-1}$ PAR under a 16/8 h day–night regimen. (**C**) Three-dimensional plot of the temperature effect on nitrate influx rate variations within a flow–force framework after logarithmic deformation in interval 100 μM to 5 mM external nitrate concentrations.

## 5. Spatial Variation of Root N Uptake Rate with Age along the Root Axes

In the 1970s, the way in which running speed deteriorates with age in athletes was established (Figure S3). Curves showed that the optimum speed improved up to age 20−30 and then deteriorated beyond age 30 years [96]. In monocarpic plant species, the question of how the rate of nitrate uptake deteriorates with root age is a major issue that has been little studied [97–100]. This question is essential because N uptake modeling has also to cope with the effects of the growth, geometry, and aging of the root system, which all affect nutrient uptake. Figure 8 shows the effect of root aging on net uptake rate expressed in root length for oilseed rape plants fed different homogeneous external nitrate concentrations (from 10 μM to 10 mM).

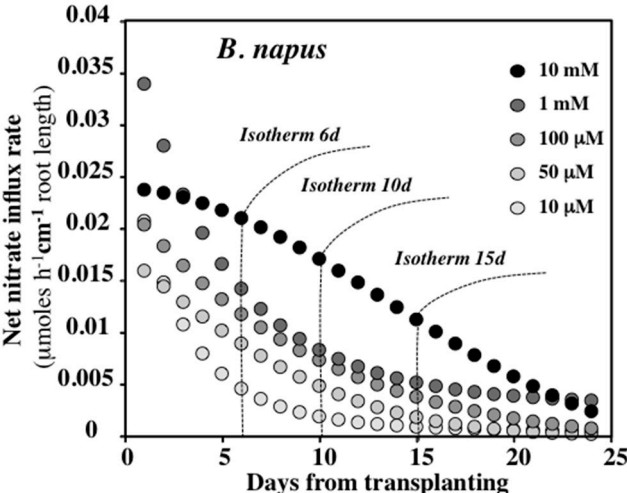

**Figure 8.** Deterioration of net uptake rate with root age in *Brassica napus* plants. The plants were grown in a continuous flow culture system with constant 10 μM, 50 μM, 100 μM, 1 mM, or 10 mM of external nitrate concentrations. The net absorption isotherm is defined as the measurement at a given temperature of the accumulation rate of an ion in the plant as a function of the external ion concentration (adapted from [100] and [19]).

The rate of net nitrate uptake gradually deteriorates from the seedling stage onwards (Figure 8), suggesting that the youngest parts of the root system are responsible for the highest rates of N absorption [100]. From these curves and for different ages of the plant, it is also possible to extract nitrate net absorption isotherms that show a continuous decrease in the $V_{maxapp}$ values (Figure 8). Increasing external concentrations of nitrate extend the root N uptake capacities without extending the duration of the N uptake period. These results confirm the previous kinetic analysis of N uptake trajectory for three levels of fertilization, where increases in the N amount taken up were due only to functional compensations in N uptake (Figure 2D). Deterioration of the N uptake rate with root age can be explained by a concomitant decrease in root respiration with age, since root respiration is involved in root construction, ion uptake, and maintenance processes [101–103].

To introduce the effect of root aging on the nitrate uptake process, we used, in the *flow–force* model [10], a synthetic parameter called Integrate Root System Age (IRSA) [97]. IRSA is defined as the sum of the average age of the root segments relative to root age at the final sampling date (maturity), and can be expressed in °C day:

$$\text{IRSA}_{ti} = \sum\nolimits_{ti=1}^{\text{maturity}} [a_{ti} \times \Delta l_{ti} / l_{\text{maturity}}] \tag{11}$$

where $a_{ti}$ is the average root age of the root segments produced from plant age $d_{I-1}$ to $d_{ti}$, $i$ is the $i$th °C day of root sampling, $\Delta l_{ti}$ represents the change in root length from $d_{ti-1}$ to $d_{ti}$, and $l_{\text{maturity}}$ is the total root length at maturity (Figure S4).

The IRSA parameter was used in the flow–force model to estimate the relative $NO_3^-$ uptake capacity of the fine-root system ($FR_{ti}$). The $FR_{ti}$ is calculated assuming a linear response between $NO_3^-$ uptake rate and root age. Thus the lowest value of the IRSA (young root segments) for each soil layer corresponds to full nitrate uptake capacity (100%) and the highest IRSA value (old root segments) was reached for absorption equal to zero (Figure S5):

$$FR_{ti} = 1 - (IRSA_{ti} / IRSA_{maturity}) \tag{12}$$

We then estimated the active root biomass (ARB$_{ti}$) involved in nitrate absorption within different soil layers throughout the growth cycle according to:

$$ARB_{ti} = 1 - (IRSA_{ti} \; / \; IRSA_{maturity}) \times Dw_{root, \, ti} \tag{13}$$

where DW$_{root,ti}$ is the root dry weight at the *i*th °C day of root sampling. When nitrate uptake rate begins to decrease after the mid-bolting stage, the active part of the roots involved in nitrate absorption represents only 4%–7% of the total fine-root system (Figure S5). Use of ARB induces a drastic reduction of the root biomass involved in nitrate uptake in the topsoil layers (L1 = 0–30 and L2 = 30–60 cm). Accordingly, the simulated values of N exported by the crop during a whole growth cycle were greatly improved in the flow–force model (Figure 9A,B).

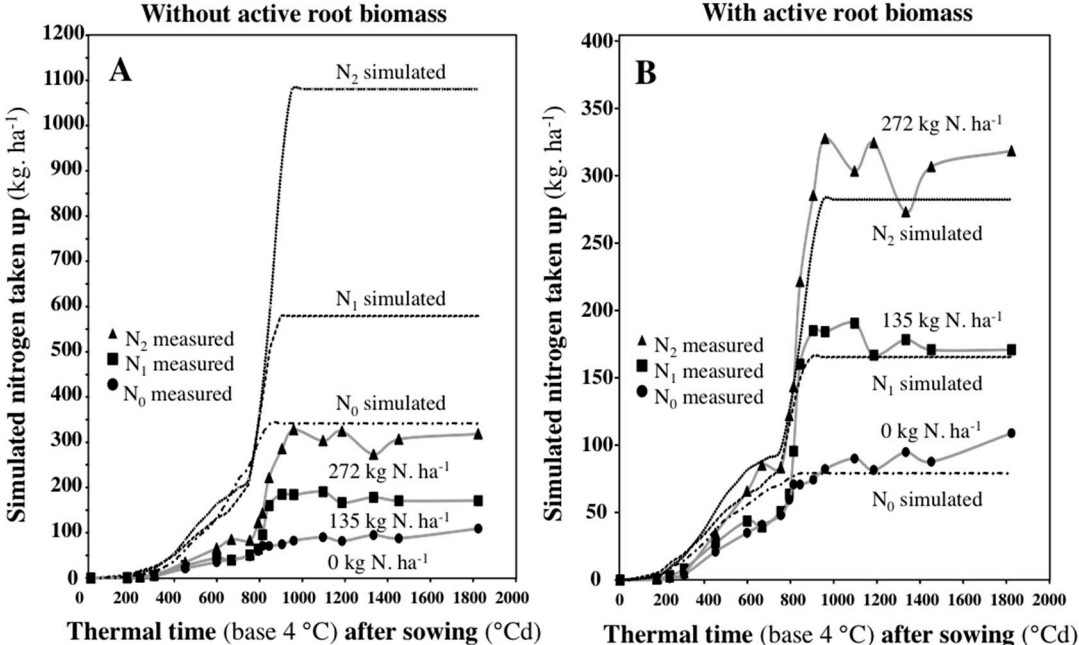

**Figure 9.** Simulated versus measured outputs of the total nitrogen taken up by a winter oil seed rape crop (*B. napus*, cultivar 'Capitol') under field conditions for three levels of N fertilization. (**A**) The simulated values are obtained without using active root biomass (ARB). (**B**) The simulated values are obtained after using ARB to estimate the active root biomass. The values simulated by the model are indicated by a dotted line and the real values measured in the field are indicated by simple symbols and a full line. Data used to run the model came from the INRA-database (https://ecosys.versailles-grignon.inra.fr/ceres_mais/base/welcome.html [15]).

The ARB parameter implicitly assumes that the active root biomass of the whole root system is mainly composed of a young fine-root system mainly located at the root tips. Introducing this functional compensation during root aging in the flow–force model suggests that the root system could evolve from an "open network" where absorption takes place all along the young root axes to a "closed network" where absorption occurs only at the tips of old roots and nearby. This behavior is very similar to that observed and modeled for water absorption [104,105] and remains consistent with diffusion and convection fluxes of nutrients in the soil. However, we have only raw physiological data to test this hypothesis [36]. To date, no study using [15]N or [13]N tracers has established the response laws involved in the activity of absorption during the aging of the root axes. Currently, only experimental protocols for measuring variations of water absorption along the roots have been proposed [106,107]. The determination of N uptake response laws along the root axes and their introduction in nitrate uptake models should significantly improve the 3-D models of N uptake. In 3-D

models, the sub-models of root development such as RootBox and SimRoot provide access to the demographics of the root axes.

As demonstrated in different split-root system experiments [108], N uptake compensations occur on root segments exposed to a high nitrate concentration. Nitrate can locally increase the expression and activity of aquaporins and nitrate transporters at the endodermis level and thereby promote nitrate translocation to the shoots [32,33,109]. In other words, we conceptually move from a "closed root ends network" to a "closed root segments network".

In phytoplankton, measurements of N uptake by a cell population also depend on the size and age of individual cells [110]. The demographic structure of juvenile, mature, and senescing cells modifies the distribution of cell sizes in a colony or community [111,112]. Although cell size is a key trait correlated with other traits such as nutrient uptake, growth and metabolism, the cell-aging component is less often considered in phytoplankton ecological models [82,110]. Growth models for cell populations structured into three cell classes (juvenile, mature, senescing) allow reformulations of logistic equations for nutrient uptake by taking into account limitations both external (limitation of a substrate) and internal (reduction of active cells caused by senescence or a fixed number of cell divisions in non-limiting substrate conditions). The flexible reformulations of the logistic growth/uptake models are alternative solutions to solve limitations encountered in the phytoplankton ecological models of nutrient uptake [12,113,114].

## 6. Conclusions

Given the close interactions between the building of root structure and root uptake properties, N uptake models cannot be improved without reconsidering the flexibility of kinetic parameters of nitrate uptake across different biological scales and also in response to environmental cues and root or cell aging. Thus, most ion absorption models are constructed from implicitly accepted, yet questionable assumptions. For example, roots are most often considered to form an open network for nitrate absorption. Similarly, it is commonly assumed that the kinetic parameters of ion absorption follow homothetic behavior at different root scales. The use of the mechanistic porter–diffusion approach developed by physiologists of marine phytoplankton and the thermodynamic flow–force interpretation of plant-based nutrient uptake isotherms could solve many problems facing modelers in these two research areas, notably by introducing temperature as an essential state variable in the thermodynamic behavior of ion absorption.

**Supplementary Materials:** The following are available online at http://www.mdpi.com/2073-4395/9/3/116/s1, Table S1: Parameters of the bilogistic N uptake model for *Brassica napus* plants growing under field conditions after three levels of fertilization. Figure S1: Variations of total length in the different soil layers of *Brassica napus* plants growing in field conditions under two levels of fertilization; Figure S2: Air temperature amplitude during the whole growth cycle of *Brassica napus* plants under field conditions; Figure S3: Deterioration of the running speed with age of men and women athletes; Figure S4: Variations of the Integrate Root System Age parameter (IRSA) during the whole growth cycle in each soil layers for two fertilization levels; Figure S5: Estimation of the active absorptive root biomass (ARB) from the active root fraction for two fertilisation levels in the topsoil layer (0–30 cm).

**Author Contributions:** Conceptualisation E.L.D., P.M. and M.-L.D. Data analyses E.L.D. and P.M. Writing-original draft preparation E.L.D. Corrections and validation E.L.D., P.M. and M.-L.D.

**Funding:** This research received no external funding.

**Acknowledgments:** 

**Conflicts of Interest:** The authors declare no conflict of interest.

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
