# Peer review of "Modelling Nitrogen Uptake in Plants and Phytoplankton: Advantages of Integrating Flexibility into the Spatial and Temporal Dynamics of Nitrate Absorption"

_agronomy, doi:10.3390/agronomy9030116_

Reviewer 1 Report

The manuscript entitled “Modelling nitrogen uptake in plants and phytoplankton: advantages of integrating flexibility into the spatio-temporal dynamic of nitrate absorption rate” reviews the concepts and models used in ion uptake kinetics spanning root and phytoplankton research. The authors outline the current limitations and challenges in modeling and propose recent solutions that could further the discipline if adopted.

Overall, I think the paper is well-written, reviews the concepts and relevant advances ion uptake kinetics clearly and provides a well-supported proposition for porter-diffusion and Flow-Force approach adoption. However, before publication I feel the following issues should be addressed:

Address Figure 9 as the point of the figure was not clear. I would suggest to focus on what a new model could look like to conclude the manuscript. Otherwise be clear that you are presenting the old model and are highlighting the limitations.

Figure resolution low and hard to read please resubmit at 300 dpi or higher.

Requires proof reading.

Additionally, I think that the following suggestions would improve the readability and understanding of the manuscript:

Line 37. Delete additional brackets.

Line 50. Rephrase as all roots classes contribute to root surface from primary roots to root hairs.

Line 62. For clarity consider using only one term after first introduction of Imax/Vmax.

Line 72. Improve clarify of sentence. eg. “…three-dimensional models (3-D) that simulate root development such as SimRoot, RootMap, SPACSYS, R-SWMS and RootBox”

Line 83. Define isotherm or put into better context for wider audience.

Line 90. Consistency in use of italics between all model names. Continues throughout manuscript.

Line 112. Table S1 missing.

Figure 1. Icons hard to identify for parts B,C,D. Figure 1A is much clearer in comparison.

Figure 1D. Right Y axis needs label. Horizontal dashed line at 0 adds to confusion. Add P1-4 to legend.

Figure 1D legend. 78+107+38 does not equal 272kg.N.ha

Figure 1D legend. Red line does not appear to be deduced from Figure 3A.

Line 139. No asterisk in Figure 1D.

Table 1. G1-4 should be renamed to P1-4 or vise verse for consistency.

Line 147. Use of “morphogenetic program” not clear, use “phenology” instead or define. Sentence is not necessary as the previous paragraph described this point well.

Line 172. Consider revising sentence for clarity.

Line 177. Lemaire and Salette, 1984 citation missing; Greewood et al., 1990 citation missing.

Line181, Equation2. Wc in text but W-b in equation.

Line 207. “the above curve” sentence continues from? Cite as Figure 2.

Line 207. Figure 2. Details 272kg N ha not 235 kg N ha.

Line 208. “closed to the optimum” optimum for what?

Line 207-209. There is an observable differences in shoot growth between N0, N1 and N2.

Line 240-242. Reference needed that supports the claim about soil heterogeneity.

Figure 3A. Data points are for fitted data and not experimental data, are the fitted points necessary? Suggest to remove curve data points or include all raw data points for Devienne-Barret et al.

Figure 3A. Any studies that support the diffusion and diffusion + convection suggestion? Should mention the other main hypothesis would be a dual transporter mechanism.

Figure 3A. Not log scale.

Figure 3B. X axis scale should be consistent with Figure 3A for clarity.

Line 249. “after” change to adapted from. Continues throughout manuscript.

Line 281-283. Reinforce these percentages are estimated.

Figure 4. Typo root axial "axe". Add approach for each study?

Figure 4. Root architectural phenes root figure, change colors according to correct root classes for clarity.

Line 411. NPF is the new nomenclature so put the NRTs in brackets instead?

Line 424. Smith et al cited paper “did not analyze data for Vmax” and instead cited key papers, reference accordingly. In addition missing Shaw et al., 2014 reference.

Line 477. Typo “tackle” tackled.

Line 508. Show et al., 2015 reference missing.

Line 508-510. Franks, 2009 is not experimental data.

Line 533. Fisken et al., 2013 reference missing.

Figure 5. Typo “Day-nigth" Day-night.

Figure 5. Legend definitions for development stages needed.

Figure 5. Include citations for data adapted in legend.

Line 591. Typo “In seventies” In the seventies.

Figure 7. Missing Figure 7A.

Figure 8. Cite data used in legend.

FigS1. N0 and N2 not defined in figure legend.

Author Response

Cover letter

Title: Modelling nitrogen uptake in plants and phytoplankton: Advantages of integrating flexibility into the spatio-temporal dynamics of nitrate absorption rate

Authors: Erwan Le Deunff, Philippe Malagoli & Marie-Laure Decau

Ref : Agronomy-406018

Dear Editor,

Please find enclosed the revised version of the manuscript entitled “Modelling nitrogen uptake in plants and phytoplankton: Advantages of integrating flexibility into the spatio-temporal dynamics of nitrate absorption rate” to be considered for publication in the Agronomy Journal.

As you will read, all minor modifications have been accounted for. More precisely, modifications have been labelled in green and red for Reviewer 1 and 2, respectively. That should make reading more convenient.

Otherwise, reading through the paper again, we noticed that this was not only a review of aggregated knowledge but it also provided new interpretation and viewpoint (Figs.  1 , 2 , 3 and 4). This is why we were wondering whether it could not be considered as an Opinion or a Viewpoint paper to better fit paper content. This is only a suggestion to your appreciation according to Agronomy Journal standard.

All authors declare being in agreement with the content of the manuscript and contributed significantly to its preparation.

Thank you in advance for your due consideration.

Yours sincerely,

Dr Erwan Le Deunff, on behalf of the authors

Reviewer 2 Report

The manuscript “Modelling nitrogen uptake in plants and phytoplankton: advantages of

integrating flexibility into the spatio-temporal dynamic of nitrate absorption rate” provides a thorough review of N uptake kinetics and how they are conventionally modelled, and contributes a synthesis of modelling advancements that account for temporal and spatial variation in uptake rates. Particularly intriguing is the potential for the proposed models to be integrated across biological scales.

Overall, the manuscript is well written, but more work is needed to eliminate errors in text. I noted some examples in my minor comments. Also, parts of the manuscript would benefit from further editing to ensure text is concise and clear.

Introduction: the importance and timeliness of this review would be better highlighted if the authors explain how this review is new compared to previous work. How is this a new synthesis of research and why is it needed now?

Please check the formatting of equations to ensure they are legible and clear.

Please improve the resolution of figures and ensure there are no red underlines below text from software spell check.

Confirm correct terminology: flow-force model or flux-force model and ensure it’s used consistently throughout manuscript.

Minor comments:

lns 97-98: To simplify this subtitle, I suggest replacing "statistical approaches" with "empirical models", and then omit at the end.

ln 99: "at the agronomic level" should be clarified/defined.

ln 103: Omit "In our case," or specify the example described and illustrated in figure 1. Throughout the review it should be clear where modelled or measured data are from.

Figure 1: Improve resolution of figure. 1.D: needs axis label for right y axis.

lns 135-136: Specify that the four phases are only shown for fertilization application level N2.

Table 1: Use consistent terminology for growth phases: P (lines 136-140 and Fig. 1) or G (table 1). ln 146: Change "deduced" to "calculated". Commas should be decimals points

ln 147 and 151: Clarify "morphogenetic program" in this context.

ln 153: Change "then the senescence process" to "then during senescence".

lns 172-175: Poor sentence structure. Please revise.

ln 183: change "status" to "statuses

ln 208: change "closed" to "close". Change "below" to "bottom".

lns 211-212: Clarify "the course in shoot morphogenetic process".

ln 213: Change "nitrogen" to "N".

lns 240-242: Specify 'in most cases' for under fertilized agronomic conditions (where presumably soil environments can become more homogenized) as certainly there can be high heterogeneity for nitrate in topsoil

lns 248-260: The caption of figure 3 is misplaced and it is unclear where it ends and the manuscript resumes. Some of this content seems like it should be in the main text with reference to the figure.

ln 250-251: Change "admit" to "is" and "the equilibrium situation in soil" to "equilibrium".

ln 270: What is "N offer"?

ln 271: insert comma after "plants".

ln 275: Omit "in plant".

ln 283: insert "soils" after "fertilized". Superscript "-1".

ln 285: Correct "analyzes" to "analysis"

ln 299: I suggest changing this subtitle. The first part is a bit strange. "What meaning should be given to" could be change to something like "How to interpret…" or “Interpreting…”.

ln 336: Remove underline for [S] in denominator.

Figure 4: Label the 'root segments' to give meaning to the dashed lines that delineate three segments. The colours of the root architectural phenes do not consistently correspond to their label. The details and reference to the Flux-force approach for roots is cut off.

ln 450-451: Specify types of phenomena. End examples in parentheses.

ln 529: State what the terms 'n' and 'h' are in these equations.

Figure 5: Correct "Day-night cycle" in 5D

ln 549: Change "the photosynthetic active radiations" to "photosynthetically active radiation".

ln 554: Specify what is meant by "all environmental conditions". This seems like an over generalization. Instead list those environmental factors.

lns 591-593: I don't think it lends to this section as the link with changes in N uptake is unclear. I suggest omitting.

Figure 7: There is no Figure 7.A, but I don't think this is needed. I suggest omitting in caption. Define isotherms in the caption.

lns 605 and 608: Not referencing Fig. 7A.

ln 623: Why "relative"? to what? Specify.

ln 631: State what the Dwroot,ti term is in eq. 12

lns 638-639: paragraph formatting error.

ln 652: Correct sentence structure: "allow to access to the demography of the root axes"

ln 652: Please improve structure: "it is not excluded that..."

ln 665-667: Correct to "other traits".

Figure 9 does not seem like a suitable figure for conclusion section. Seems more appropriate in introduction.

ln 685: Please edit and improve "it is admitted".

Author Response

Cover letter

Title: Modelling nitrogen uptake in plants and phytoplankton: Advantages of integrating flexibility into the spatio-temporal dynamics of nitrate absorption rate

Authors: Erwan Le Deunff, Philippe Malagoli & Marie-Laure Decau

Ref : Agronomy-406018

Dear Editor,

Please find enclosed the revised version of the manuscript entitled “Modelling nitrogen uptake in plants and phytoplankton: Advantages of integrating flexibility into the spatio-temporal dynamics of nitrate absorption rate” to be considered for publication in the Agronomy Journal.

As you will read, all minor modifications have been accounted for. More precisely, modifications have been labelled in green and red for Reviewer 1 and 2, respectively. That should make reading more convenient.

Otherwise, reading through the paper again, we noticed that this was not only a review of aggregated knowledge but it also provided new interpretation and viewpoint (Figs.  1 , 2 , 3 and 4 and Tables SI, I and II). This is why we were wondering whether it could not be considered as an Opinion or a Viewpoint paper to better fit paper content. This is only a suggestion to your appreciation according to Agronomy Journal standard.

All authors declare being in agreement with the content of the manuscript and contributed significantly to its preparation.

Thank you in advance for your due consideration.

Yours sincerely,

Dr Erwan Le Deunff, on behalf of the authors